# Infrared and Visible Image Fusion for Highlighting Salient Targets in the Night Scene

**DOI:** 10.3390/e24121759

**Published:** 2022-11-30

**Authors:** Weida Zhan, Jiale Wang, Yichun Jiang, Yu Chen, Tingyuan Zheng, Yang Hong

**Affiliations:** National Demonstration Center for Experimental Electrical, School of Electronic and Information Engineering, Changchun University of Science and Technology, Changchun 130022, China

**Keywords:** image fusion, highlighting salient targets, infrared images, deep learning

## Abstract

The goal of infrared and visible image fusion in the night scene is to generate a fused image containing salient targets and rich textural details. However, the existing image fusion methods fail to take the unevenness of nighttime luminance into account. To address the above issue, an infrared and visible image fusion method for highlighting salient targets in the night scene is proposed. First of all, a global attention module is designed, which rescales the weights of different channels after capturing global contextual information. Second, the loss function is divided into the foreground loss and the background loss, forcing the fused image to retain rich texture details while highlighting the salient targets. Finally, a luminance estimation function is introduced to obtain the trade-off control parameters of the foreground loss function based on the nighttime luminance. It can effectively highlight salient targets by retaining the foreground information from the source images. Compared with other advanced methods, the experimental results adequately demonstrate the excellent fusion performance and generalization of the proposed method.

## 1. Introduction

Image fusion fuses multimodal images of the same scene to produce a fused image with richer information and clearer details [1,2]. Infrared and visible image fusion, as an indispensable part in image fusion direction, has been applied in all walks of life [3,4,5]. The fused image can be utilized as a preprocessing unit for subsequent high-level visual tasks such as object detection, object tracking and semantic segmentation. As shown in Figure 1, the salient target of a human is mainly concentrated in the thermal radiation information of the infrared image in the night scene. In addition, the uneven luminance of the night scene results in the uneven distribution of texture details in the infrared and visible images. Therefore, the fused image can effectively retain texture details while highlighting salient targets [6].

Constrained by the local receptive field of the single convolution layer, U2Fusion [7], IFCNN [8] and RXDNFuse [9] capture global contextual information by stacking plenty of convolution layers, resulting in a huge computation workload. GANMcC [10] uses the infrared and visible images as the inputs of the discriminator. It can retain the information in the source images but fails to highlight the salient targets. STDFusion [11] designs the loss function by considering only the infrared foreground and the visible background, possibly leading to the loss of some important information. Therefore, there are no comprehensive loss functions to supervise the training of the network.

To solve these problems, this paper proposes an infrared and visible image fusion method for highlighting salient targets in the night scene. The major contributions of this paper are as follows:To reduce the computational burden caused by stacking convolution layers, a global attention module is added to the image fusion network. This module rescales the weights of different channels after capturing global contextual information. Assigning higher weight for the important channels makes the salient targets more prominent;In order to improve the comprehensiveness of the loss function, the loss function of the proposed method is divided into the foreground loss and the background loss, forcing the fused image to contain more prominent salient targets and richer texture details;To address the problem of the uneven luminance in the night scene, a luminance estimation function is introduced into the training process. It outputs the trade-off control parameters of the foreground loss function based on the nighttime luminance, which helps to retain the foreground information from source images more reasonably.

The remainder of this paper is arranged as follows. In Section 2, the reviews about infrared and visible image fusion methods and attention mechanism are presented. In Section 3, the loss function and structure of the proposed network are elaborated. The effectiveness and generalization of the proposed method are demonstrated in Section 4. Finally, the conclusion is expressed in Section 5.

## 2. Related Work

In this section, a literature review of infrared and visible image fusion methods and attention mechanisms are presented.

### 2.1. Infrared and Visible Image Fusion

Traditional image fusion methods usually use a fixed mathematical transformation to extract image features, possibly leading to a weak feature representation. Moreover, these methods do not take modal differences between multimodal images into account [12]. Deep learning-based image fusion methods, with a powerful representation capability, transform the fusion process into training different depths of neural networks.

Over the last few years, convolutional neural network-based image fusion methods have gained notable success. Xu et al. [13] adopted a dense connection structure to extract image features and added multiple jump connections between the dense blocks to increase the information flow. Tang et al. [14] proposed an image fusion network based on illumination perception, which could adaptively maintain the intensity distribution of salient targets according to light distribution.

Due to the capability of unsupervised distribution estimation, a GAN-based network is fit for image fusion tasks. Based on the former research [15], Ma et al. [16] proposed a GAN-based image fusion network with two discriminators. The infrared and visible images were adopted as the inputs of two discriminators, respectively, forcing the fused result to contain the information from both source images. Hou et al. [17] divided the input of generators into a gradient path and a contrast path and designed an adjusted image as the ground truth for the discriminator.

As the research progressed, a number of in-depth methods based on auto-encoder were proposed. Li et al. [18] adopted an encoder with nest connection structure to fully exploit the deep features, and a hybrid attention module with channel and spatial attention was added to the fusion layer. Li et al. [19] proposed a special fusion layer with four modified residual fusion modules. The shallow and deep fusion modules were used to extract texture details and retain the significant structure, respectively. Particularly, the coding and fusion networks were trained in stages.

Although excellent fusion results can be obtained using the above methods, they still fail to effectively highlight salient targets in the fused image. The loss function of the proposed method is divided into the foreground loss and the background loss, which helps to more effectively extract and retain texture details while highlighting the salient targets.

### 2.2. Attention Mechanism

The attention mechanism is of great significance in the human visual system. When faced with a complex scene, the human eye tends to focus more on the important targets and ignore the irrelevant information, which helps to analyze complex scenes quickly and efficiently [20]. The attention mechanism enables selective focus on important parts by reassigning weights to input sequences [21].

The attention mechanism has achieved great success in some fields, including target detection [22], image enhancement [23], semantic segmentation [24] and emotion recognition [25]. Recently, many scholars have applied the attention mechanism to image fusion tasks. For instance, Li et al. [26] introduced a multiscale channel attention network into the generator and discriminator, respectively, to help the network focus more on salient regions in the image. Li et al. [27] argued that both channel and spatial dimensions contain rich attention information, therefore, the channel and spatial attention modules were combined for further feature extraction.

To break through the limitation of the local receptive field of the single convolution layer, Wang et al. [28] proposed a non-local attention module, which could model global contextual relationships to capture feature dependencies over long distance. However, multiple matrix multiplications were applied in this module, leading to inefficient learning of the network. In order to reduce the computational complexity, many scholars have proposed some variants based on the non-local attention module [29,30].

Despite the fact that the non-local attention module is effective in capturing global contextual information, it is unable to model the correlations between channels. Therefore, a global attention module was designed, which rescales the weights of different channels after modeling global contextual relationships.

## 3. The Proposed Method

In this section, the problem formulation and the core ideas are discussed. Then, the design of the loss function is presented. Finally, the structure of the image fusion network is described in detail.

### 3.1. Problem Formulation

A fused image can compensate for the lack of information from a single sensor [31]. The ideal result of the fused image is to retain rich texture details while highlighting the salient targets. As shown in Figure 2, in the training process, the source images were fed into the image fusion network which output the fused image. The luminance estimation function evaluated the luminance of all visible images in the training dataset and output the trade-off control parameters of the foreground loss function. Under the surveillance of the loss function, the fusion performance was improved.

In the testing process, the final fused image was obtained by feeding the source images into the image fusion network directly without any preprocessing. The design of the loss function and the image fusion network are described in detail as follows.

### 3.2. Loss Function

To obtain the fused result with salient targets and rich texture details, the loss function was divided into the foreground loss and the background loss. The foreground with salient targets and the background containing rich texture details were split from the images [17].

As shown in Figure 3, humans were defined as the salient targets in this paper. First of all, the semantic segmentation results of the infrared image were acquired using Mask R-CNN [32]. The mask for humans was kept and then set as a binary image. As shown in Figure 3c,d, the foreground mask and the background mask were obtained. The source images were split into the foreground and background using the above masks [17]. It is worth mentioning that IR-Fore, VI-Fore, IR-Back and VI-Back represent the foreground and background of the infrared and visible image, respectively.

In Figure 3, most of the salient target information is concentrated in the infrared foreground in the night scene. However, the visible foreground also contained some salient target information. The lower the nighttime luminance, the less salient target information was contained in the visible foreground. Therefore, a luminance estimation function was introduced to evaluate the nighttime luminance. The luminance estimation function is formulated in Equations (1) and (2).
(1)$$\beta_{2}=\frac{1}{N}\overset{N}{\underset{n=1}{\sum }}\left(1-\log\left(1+e^{-p_{n}}\right)\right)$$
(2)$$\beta_{2}=1-\beta_{1}$$
where $\beta_{1}$ and $\beta_{2}$ represent the trade-off control parameters of the foreground loss function. *N* denotes the number of visible images in the training dataset. $p_{n}$ denotes the normalized average grey level value of the nth input image. According to the analysis of Equation (1), the lower the nighttime luminance, the smaller the value of $\beta_{2}$.

The foreground loss function contains the foreground pixel loss $L_{pixel-fore}$ and the foreground gradient loss $L_{grad-fore}$, which are defined in Equations (3) and (4).
(3)$$L_{pixel-fore}=\frac{\beta_{1}\left|I_{m}\cdot (I_{f}-I_{ir})\right|+\beta_{2}\left|I_{m}\cdot (I_{f}-I_{vi})\right|}{HW}$$
(4)$$L_{grad-fore}=\frac{\beta_{1}\left|I_{m}\cdot (\nabla I_{f}-\nabla I_{ir})\right|+\beta_{2}\left|I_{m}\cdot (\nabla I_{f}-\nabla I_{vi})\right|}{HW}$$
where $I_{m}$, $I_{f}$, $I_{ir}$ and $I_{vi}$ represent the foreground mask, the fused image, the infrared and the visible image, respectively. H and W denote the height and width of the image. $\left|\cdot \right|$ indicates the mean absolute error. ∇ and · denote the gradient operator and element multiplication operator, respectively. The fused results for different values of $\beta_{1}$ and $\beta_{2}$ are analyzed in Section 4.

The background luminance was uneven in the night scene, which lead to an uneven distribution of texture details in the source images as well. When the luminance was high, most of the texture details were concentrated in the visible background. As the luminance decreased, most of the texture details were concentrated in the infrared background. Therefore, for the background loss function, the highest pixel and gradient intensity distribution of the source images were chosen. Equations (5) and (6) for the background pixel loss $L_{pixel-back}$ and the background gradient loss $L_{grad-back}$ are shown as follows.
(5)$$L_{pixel-back}=\frac{\left|\left(1-I_{m}\right)\cdot (I_{f}-max(I_{ir},I_{vi}))\right|}{HW}$$
(6)$$L_{grad-back}=\frac{\left|\left(1-I_{m}\right)\cdot \left(\left|\nabla I_{f}\right|-\max\left(\left|\nabla I_{ir}\right|,\left|\nabla I_{vi}\right|\right)\right)\right|}{HW}$$
where $\left(1-I_{mask}\right)$ represents the background mask and max (∙) denotes the element of the max selection. The total loss function L is defined in Equation (7).
(7)$$L=\alpha_{1}(L_{pixel-fore}+L_{grad-fore})+\alpha_{2}(L_{pixel-back}+L_{grad-back})$$
where $\alpha_{1}$ + $\alpha_{2}$ = 10. The fused results for different values of $\alpha_{1}$ and $\alpha_{2}$ are analyzed specifically in Section 4.

### 3.3. Network Structure

As shown in Figure 4, the image fusion network consisted of three parts, including the feature extraction module, global attention module and image reconstruction module. First, the feature extraction module extracted the features of the source images. Then, the global attention module aggregated the global contextual information and assigned higher weight to the important channels. Finally, the image reconstruction module output the final result by reconstructing all the feature maps concatenated in the channel dimension.

#### 3.3.1. Feature Extraction Module

Since different types of source images had modal differences, the input was split into two paths. The feature extraction module for each path was designed with the same structure and respective parameters.

The feature extraction module was composed of a shallow and deep feature extraction module. In particular, the shallow feature extraction module contained two convolution layers and an activation function layer. There were two convolution blocks in the deep feature extraction module which further extracted the image features adequately. Furthermore, each block contained a convolution layer and an activation function layer. A dense connection structure was applied between each block to ensure the full utilization of information. It is necessary to note that the padding of each convolution layer was set to *SAME* and the stride was set to one. The parameter settings of the feature extraction module are shown in Table 1.

#### 3.3.2. Global Attention Module

Two global attention modules were introduced after the shallow and deep feature extraction modules. First, the global context was modeled to capture long-range feature dependencies. Later, the correlation between each channel was built through a fully connected layer. The structure of the global attention module is shown in Figure 5.

For the input feature $\Phi^{I}\in R^{C\times H\times W}$, it was first reshaped and transposed to $\Phi^{X}\in R^{C\times HW}$. Then, $\Phi^{I}$ was reshaped to $\Phi^{Y}\in R^{HW\times 1\times 1}\;$ after a convolution layer. A softmax operation was applied to calculate the attention map $A\in R^{C\times 1\times 1}$, which is defined by:(8)$$A=\frac{\exp\left(\Phi^{Y}\right)}{\sum_{i=1}^{HW}\exp\left(\Phi^{Y}\right)}$$
where $i\in \left[1,HW\right]$. Subsequently, a matrix multiplication operation between A and $\Phi^{X}$ was performed, obtaining the global attention map ${\;\Phi }^{G}\in R^{C\times 1\times 1}$, which is defined by:(9)$$\Phi^{G}=A\times \Phi^{X}$$
where × denotes the matrix multiplication operation. After obtaining the global context dependencies, the channel relationship was modeled with a fully connected layer. Then, a softmax operation was applied to obtain the global attention map $G\in R^{C\times 1\times 1}$, which is defined by:(10)$$G=\;\sigma (f_{c}\left(W\right))\;$$
where $f_{c}\left(\cdot \right)$ denotes the fully connected layer, W is the parameter of the fully connected layer and σ (∙) denotes the Sigmoid operation. Finally, the final output feature map $\Phi^{O}\in R^{C\times H\times W}$ was obtained by applying the corresponding pixel multiplication operation between *G* and $\Phi^{I}$. The final output feature map $\Phi^{O}$ is formulated as follows.
(11)$$\Phi^{O}=G\times \Phi^{I}\;$$
where × denotes a corresponding pixel multiplication operation.

#### 3.3.3. Image Reconstruction Module

The image reconstruction module reconstructed the whole feature maps concatenated in the channel dimension and output the final result. There were four convolution blocks in this module. In particular, each block contained a convolution layer and an activation function layer. Moreover, the padding of each convolution layer was set to *SAME* and the stride was set to one. The image fusion network did not change the size of the input. The parameter settings of the image reconstruction module are shown in Table 2.

## 4. Experiments

The effectiveness and generalization of the proposed method are demonstrated in this section. First of all, the experimental details are presented. Then, the experimental results of two datasets are discussed in detail. Finally, ablation experiments concerning the global attention module, the feature extraction module and the trade-off control parameter are presented.

### 4.1. Experimental Details

#### 4.1.1. Training Dataset

The MSRS dataset [14] was chosen to train the network model. This dataset was a mixture of daytime and nighttime road-related scenes, containing various elements such as pedestrians, vehicles, buildings, etc. One hundred and eighty-eight pairs of the nighttime images were selected from the dataset. The foreground mask and the background mask were obtained through the modified semantic segmentation network Mask R-CNN [32]. Each selected image was cropped into image patches with the size of 120 × 120 by setting the stride to 64, resulting in 13,160 patch pairs for the network training. In the testing process, 26 and 16 pairs of images were chosen from the MSRS dataset and the M^3^FD dataset, respectively, for comparison experiments. Note that during the testing process, the input did not require any preprocessing.

#### 4.1.2. Training Details

The network model was implemented on the TensorFlow platform, and Adam was used as the optimization solver. The batch size and epoch were set to 32 and 30, respectively. In addition, the learning rate was set to 10 × 10^−3^. Particularly, we set $\alpha_{1}=4$, $\alpha_{2}=6$, $\beta_{1}=0.66$ and $\beta_{2}=0.34$.

This paper evaluates the quality of fused results by two means containing qualitative and quantitative comparisons. Six compared methods with the code and default parameters were chosen, including GTF [33], DenseFuse [34], U2Fusion [7], GANMcC [10], RFN-Nest [19] and STDFusion [11]. Six typical evaluation metrics were selected including entropy (EN) [35], mutual information (MI) [36], standard deviation (SD) [37], spatial frequency (SF) [38], visual information fidelity (VIF) [39] and quality of images (Q^ab/f^) [40]. It is worth mentioning that the larger the above evaluation metrics, the better the fused result.

### 4.2. Results of the MSRS Dataset

For assessing the fusion performance of the proposed method in a comprehensive manner, the MSRS dataset was chosen for comparison with six state-of-the-art methods.

#### 4.2.1. Qualitative Results

For visually comparing the fusion performance of different methods, four pairs of images from the MSRS dataset were chosen for qualitative comparison. Notably, in Figure 6, Figure 7, Figure 8 and Figure 9, red and green boxes were used to mark the salient target and texture details, respectively. All methods achieved excellent performance in retaining both salient targets and texture details. Specifically, as shown in Figure 6, DenseFuse [34], U2Fusion [7], GANMcC [10] and STDFusion [11] were able to highlight the salient targets well, but the texture details of the building’s windows in the background were not clear. GTF [33] and RFN-Nest [19] retained the thermal radiation information well, but the edges of the salient targets were blurred.

For the other three scenes, the proposed method effectively highlighted the salient targets. In addition, it also had some advantages in retaining texture details in the background. To be specific, our fused result had the sharpest detail of the tree branches in Figure 7. As shown in Figure 8, the outline of the car wheels was clear. Moreover, the tree trunk could be clearly distinguished from its surroundings in Figure 9. The above results indicated that the proposed method had better capabilities in highlighting salient targets and preserving rich texture details.

#### 4.2.2. Quantitative Results

For further evaluating the effectiveness of different methods, 26 pairs of images from the MSRS dataset were chosen for quantitative comparison. As shown in Figure 10 and Table 3, the proposed method had a remarkable advantage in five metrics, including EN, MI, SD, SF and Q^ab/f^. The value of the VIF for the proposed method was second to the value for RFN-Nest by a narrow margin. Specifically, the best results for EN and MI showed that the fused results contained a whole wealth of information transferred from the source images. The maximum values of the SD and SF suggested that our fused image had a relatively high contrast ratio. Even if the value of the VIF was not the best, the fused result still effectively highlighted the salient targets and preserved rich texture details.

### 4.3. Results of the M^3^FD Dataset

For further evaluating the generalization of the proposed method, the M^3^FD dataset was used to test the network model trained using the MSRS dataset.

#### 4.3.1. Qualitative Results

A pair of images was chosen from the M^3^FD dataset for qualitative comparison. As shown in Figure 11, all methods retained the infrared thermal radiation information as well as the texture details. Specifically, the fused images of the proposed method were rich in thermal radiation information. Both near and distant targets could be easily distinguished from the background. Compared with other methods, the textural details of the proposed method were much clearer and more suitable for visual observation, for instance the detail of the building in Figure 11.

The excellent results demonstrate the generalization of the proposed method. To sum up, the fused results were effective in highlighting salient targets when applied in the night scene, and the texture details were clearly visible.

#### 4.3.2. Quantitative Results

Sixteen pairs of images from the M^3^FD dataset were chosen for quantitative comparison. As shown in Figure 12 and Table 4, the proposed method obtained the maximum mean values for four evaluation metrics including EN, MI, VIF and Q^ab/f^. This indicated that our fused images were rich in information and were more suitable for observation with the human visual system. Although our proposed method did not achieve the best results for the SD and SF, the fused images still had a sufficiently high contrast. The above experimental results adequately prove the excellent fusion performance and outstanding generalization.

### 4.4. Ablation Experiments

#### 4.4.1. Global Attention Module

The perceptual field of the single convolution layer is limited, so the contextual information can only be captured by stacking plenty of convolution layers. To address the huge computational effort, the global attention module was designed. This module modeled the global contextual relationships and then rescaled the weights of different channels to make salient targets more prominent. For confirming the usefulness of the global attention module, three ablation experiments were designed, involving removing the first and the second global attention module, respectively, and removing both global attention modules.

One hundred pairs of images from the M^3^FD dataset were chosen for qualitative and quantitative analyses. In Figure 13, for the fused result with the single global attention module, the salient targets were relatively blurred. The fused result without the global attention module performed the worst in highlighting salient targets. In Table 5, the maximum values for five evaluation metrics were achieved when two global attention modules were retained. Therefore, the global attention module assigned higher weights to the significant channels, resulting in the effective highlighting of the salient targets.

In order to further verify its advancement, the global attention module was replaced with SENet [41] and ECANet [42]. As shown in Figure 13, all global attention modules effectively highlighted the salient targets. Therefore, it was difficult to judge the superiority of these modules with human visual inspection. Therefore, 100 pairs of images from the M^3^FD dataset were utilized for further quantitative comparison. In Table 6, all evaluation metrics of the proposed module achieved maximum values. Therefore, all results confirmed its advancement.

#### 4.4.2. Feature Extraction Module

The global attention module effectively avoided the problem of high computational complexity caused by stacking convolution layers. To further verify the effect of the number of convolution blocks in the deep feature extraction module, two ablation experiments were designed. Three or four convolution blocks were adopted in the deep feature extraction module. The output channels of the third and fourth convolution layer were set up to 96 and 128, respectively. The other parameters were set up as before. One hundred pairs of images from the M^3^FD dataset were chosen for quantitative analysis. As shown in Table 7, the proposed method achieved the maximum values for four evaluation metrics including EN, MI, SD and VIF. Note that the evaluation metrics did not rise with the increase in convolution blocks, which meant that the network was possibly overfitted. Therefore, only two convolution blocks were adopted in the deep feature extraction module.

#### 4.4.3. Trade-Off Control Parameters

For retaining the salient targets and rich texture details at the same time, the loss function was divided into the foreground loss and the background loss. The trade-off control parameters of the foreground loss were $\alpha_{1}$ and $\alpha_{2}$, where $\alpha_{1}+\alpha_{2}$ = 10. Fixing the other parameters, we set $\alpha_{1}=1,2,\ldots ,9$. One hundred pairs of images from the M^3^FD dataset were selected for analysis. The variation trends in the evaluation metrics under different values of $\alpha_{1}$ can be observed in Figure 14. In Table 8, the best results were obtained for all evaluation metrics except for Q^ab/f^ when $\alpha_{1}$ was set to four.

By evaluating the luminance of all visible images in the training dataset, the luminance estimation function output the trade-off control parameters $\beta_{1}$ and $\beta_{2}$, where $\beta_{1}+\beta_{2}=1$. Fixing the other parameters in the network, we set $\beta_{1}=0.1,0.2,\ldots ,0.9$. One hundred pairs of images from the M^3^FD dataset were selected for qualitative and quantitative analyses. In Figure 15, the variation trends in the evaluation metrics under the different values of $\beta_{1}$ are shown. As shown in Table 9, our proposed method acquired the best results for four evaluation metrics including EN, MI, SD and VIF.

In order to further verify the validity of the luminance estimation function, the experiment was continued with a reduction in the step length. We set $\beta_{1}=0.61,0.62,\ldots ,0.69$ and kept the other parameters in the network unchanged.

One hundred pairs of images from the M^3^FD dataset were chosen for quantitative analysis using the above 10 sets of ablation experiments. In Figure 16, the variation trends in the evaluation metrics under the different values of $\beta_{1}$ are shown. As shown in Table 10, our proposed method acquired the best results for four evaluation metrics including EN, SD, SF and VIF. In summary, the luminance estimation function could effectively evaluate the luminance of the night scene and output a reasonable value of β.

As shown in Figure 17, the salient targets were very prominent and the details of the building were very distinct when $\alpha_{1}$ and $\beta_{1}$ were set to 4 and 0.66, respectively. All the above experiments adequately prove the rationality of the parameter settings in this paper.

## 5. Conclusions

This paper proposed an infrared and visible image fusion method for highlighting salient targets in the night scene. First, a global attention module was designed, which rescaled the weights of different channels after capturing the global contextual information. Second, the loss function was divided into the foreground loss and the background loss, forcing the fused image to retain rich texture details while highlighting the salient targets. Moreover, a luminance estimation function was introduced to obtain the trade-off control parameters of the foreground loss function based on the nighttime luminance. It effectively highlighted salient targets by retaining foreground information from the source images. In summary, the effectiveness and generalization of the proposed method was demonstrated by comparing with other advanced methods.

## Figures and Tables

**Figure 1 entropy-24-01759-f001:**
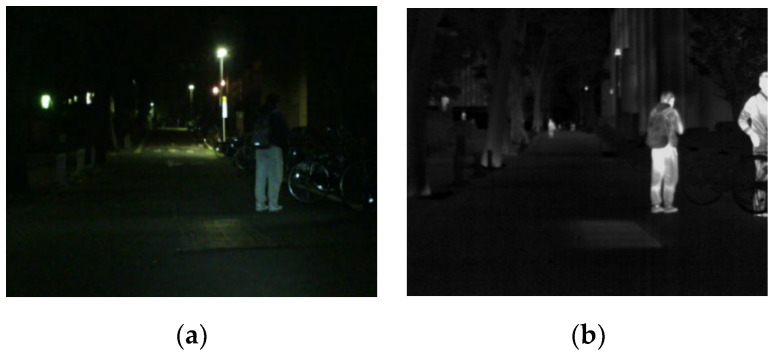
The infrared and visible images in the night scene. (**a**) Visible. (**b**) Infrared.

**Figure 2 entropy-24-01759-f002:**
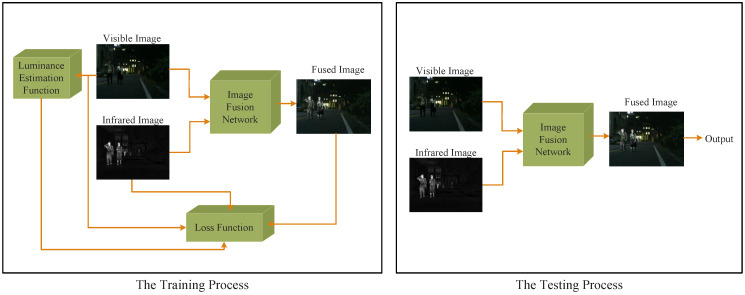
The entire procedure of the proposed method.

**Figure 3 entropy-24-01759-f003:**
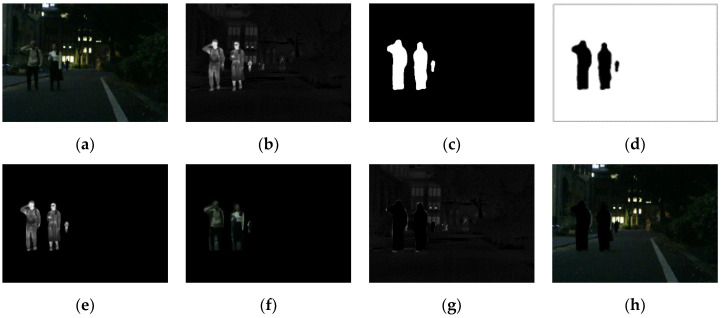
The source images, masks, foreground and background. (**a**) Visible. (**b**) Infrared. (**c**) Mask-Fore. (**d**) Mask-Back. (**e**) IR-Fore. (**f**) VI-Fore. (**g**) IR-Back. (**h**) VI-Back.

**Figure 4 entropy-24-01759-f004:**
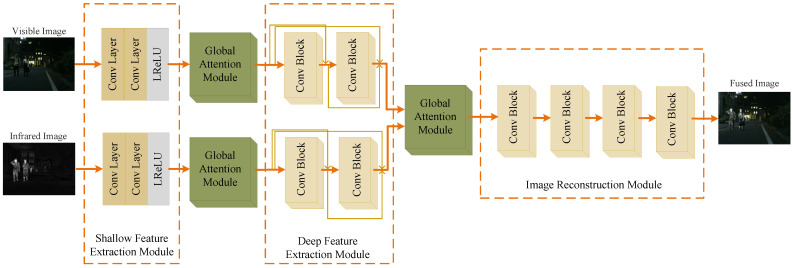
The structure of the image fusion network.

**Figure 5 entropy-24-01759-f005:**
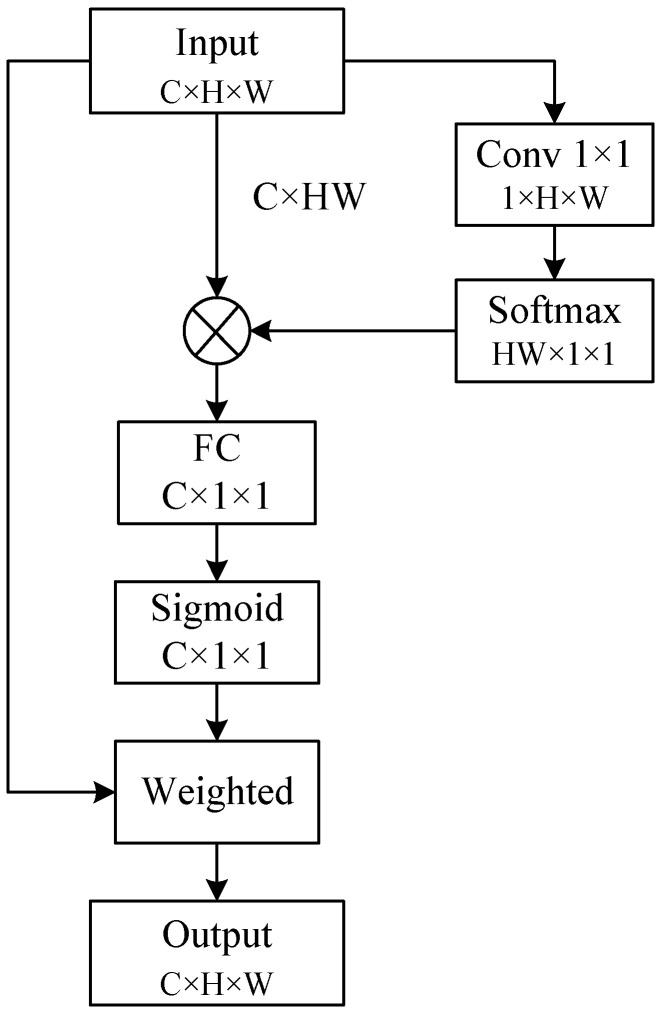
Structure of the global attention module.

**Figure 6 entropy-24-01759-f006:**
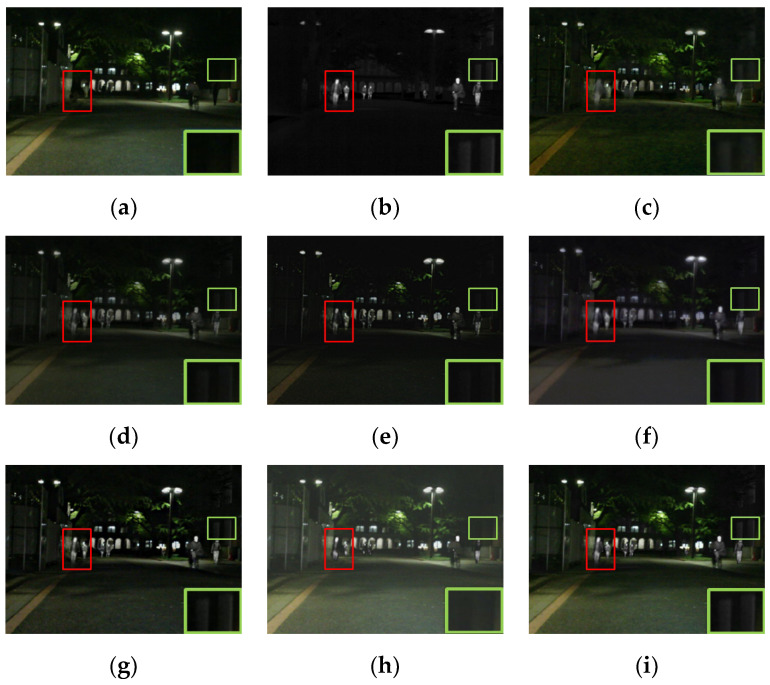
The qualitative results of the first scene of the MSRS dataset. (**a**) Visible. (**b**) Infrared. (**c**) GTF. (**d**) DenseFuse. (**e**) U2Fusion. (**f**) GANMcC. (**g**) RFN-Nest. (**h**) STDFusion. (**i**) Ours.

**Figure 7 entropy-24-01759-f007:**
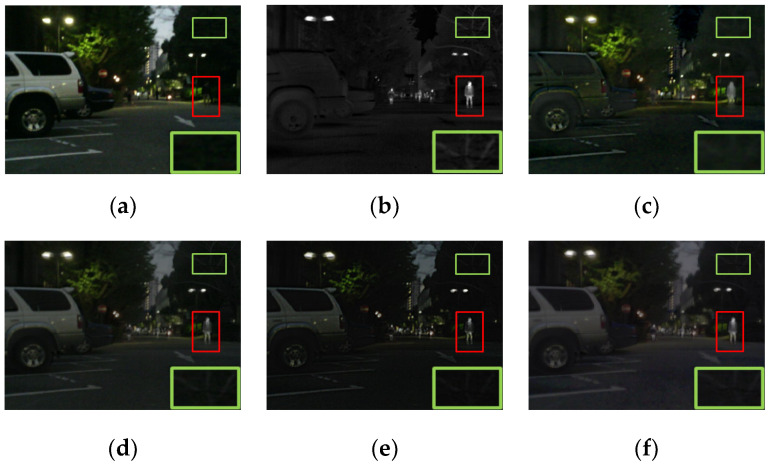
The qualitative results of the second scene of the MSRS dataset. (**a**) Visible. (**b**) Infrared. (**c**) GTF. (**d**) DenseFuse. (**e**) U2Fusion. (**f**) GANMcC. (**g**) RFN-Nest. (**h**) STDFusion. (**i**) Ours.

**Figure 8 entropy-24-01759-f008:**
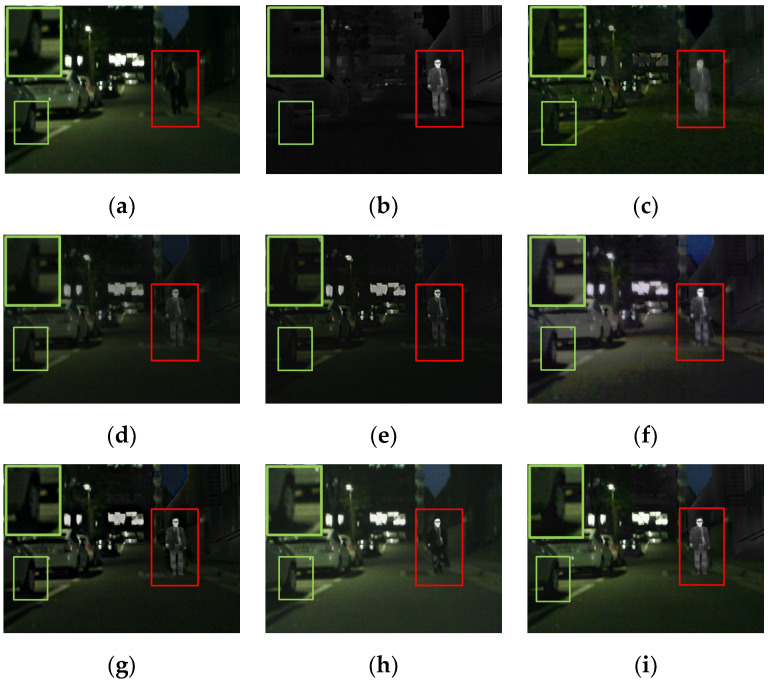
The qualitative results of the third scene of the MSRS dataset. (**a**) Visible. (**b**) Infrared. (**c**) GTF. (**d**) DenseFuse. (**e**) U2Fusion. (**f**) GANMcC. (**g**) RFN-Nest. (**h**) STDFusion. (**i**) Ours.

**Figure 9 entropy-24-01759-f009:**
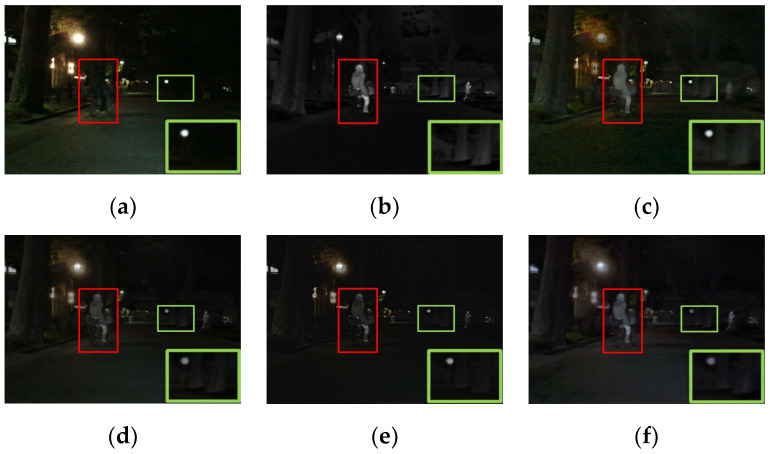
The qualitative results of the fourth scene of the MSRS dataset. (**a**) Visible. (**b**) Infrared. (**c**) GTF. (**d**) DenseFuse. (**e**) U2Fusion. (**f**) GANMcC. (**g**) RFN-Nest. (**h**) STDFusion. (**i**) Ours.

**Figure 10 entropy-24-01759-f010:**
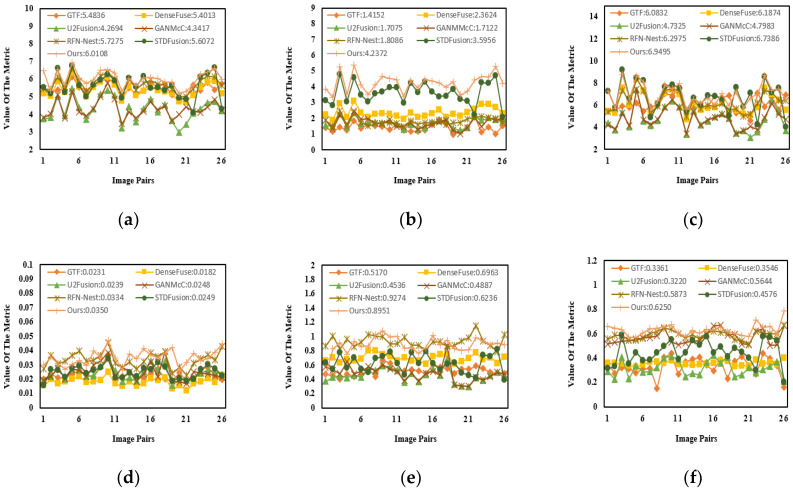
The quantitative results of the MSRS dataset. (**a**) EN. (**b**) MI. (**c**) SD. (**d**) SF. (**e**) VIF. (**f**) Q^ab/f^.

**Figure 11 entropy-24-01759-f011:**
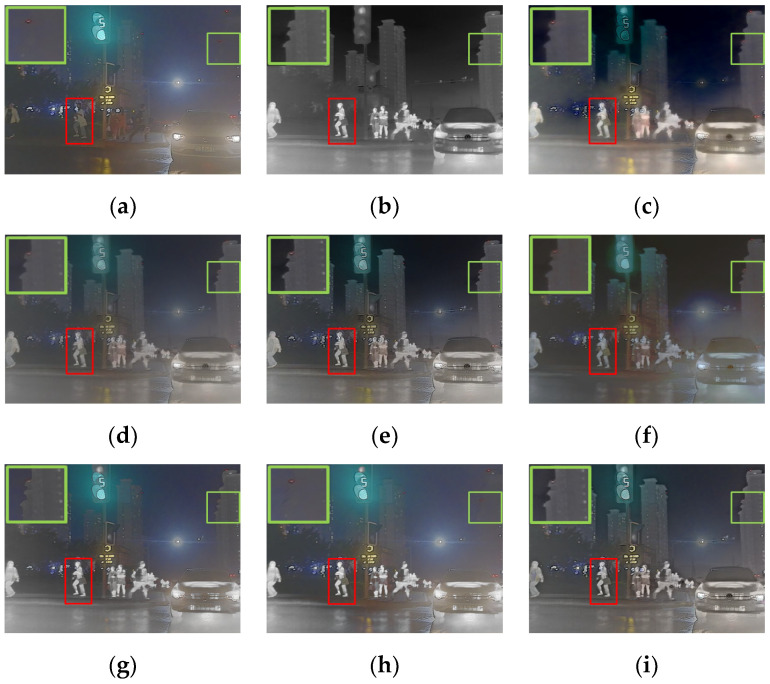
The qualitative results of the M^3^FD dataset. (**a**) Visible. (**b**) Infrared. (**c**) GTF. (**d**) DenseFuse. (**e**) U2Fusion. (**f**) GANMcC. (**g**) RFN-Nest. (**h**) STDFusion. (**i**) Ours.

**Figure 12 entropy-24-01759-f012:**
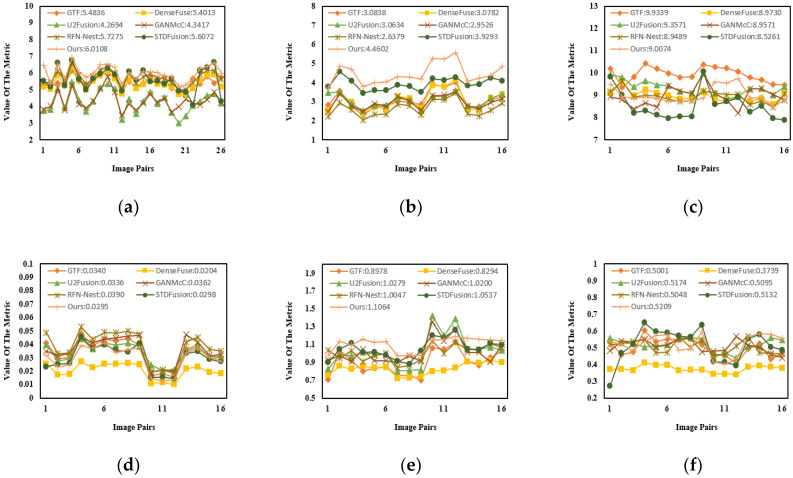
The quantitative results of the M^3^FD dataset. (**a**) EN. (**b**) MI. (**c**) SD. (**d**) SF. (**e**) VIF. (**f**) Q^ab/f^.

**Figure 13 entropy-24-01759-f013:**
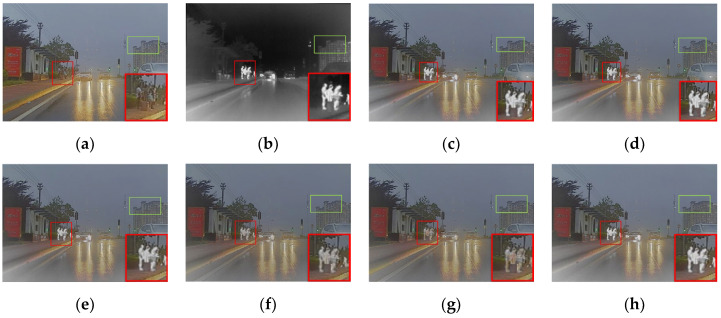
The qualitative results of ablation experiments concerning the global attention module. (**a**) Visible. (**b**) Infrared. (**c**) Replacing attention module with SENet. (**d**) Replacing attention module with ECANet. (**e**) Without the 1st attention module. (**f**) Without the 2nd attention module. (**g**) Without both attention modules. (**h**) Ours.

**Figure 14 entropy-24-01759-f014:**
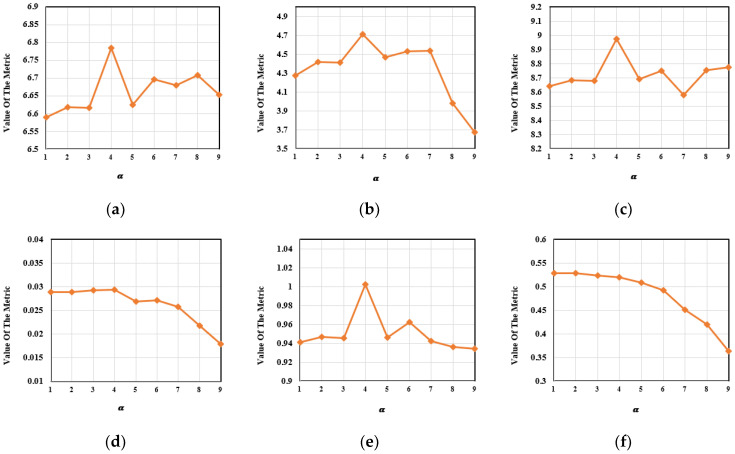
The variation trends in the metrics for the trade-off control parameter α. (**a**) EN. (**b**) MI. (**c**) SD. (**d**) SF. (**e**) VIF. (**f**) Q^ab/f^.

**Figure 15 entropy-24-01759-f015:**
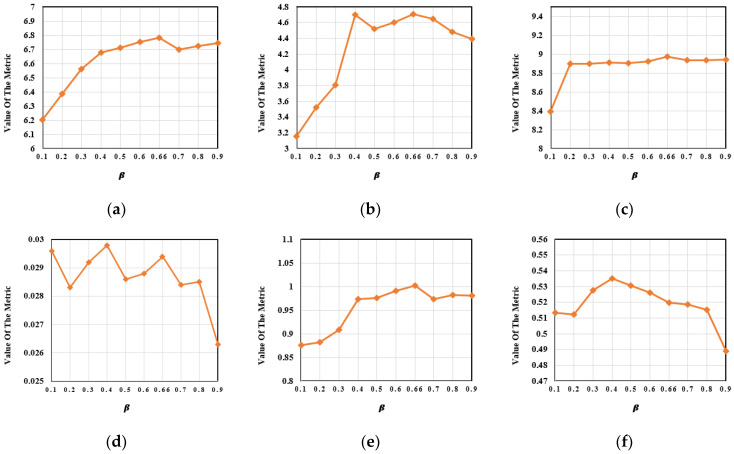
The variation trends in the metrics for the trade-off control parameter *β*. (**a**) EN. (**b**) MI. (**c**) SD. (**d**) SF. (**e**) VIF. (**f**) Q^ab/f^.

**Figure 16 entropy-24-01759-f016:**
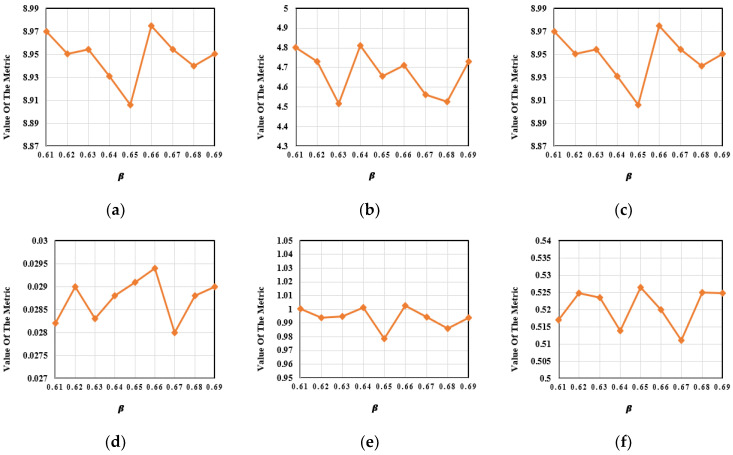
The further ablation experimental results for the trade-off control parameter *β*. (**a**) EN. (**b**) MI. (**c**) SD. (**d**) SF. (**e**) VIF. (**f**) Q^ab/f^.

**Figure 17 entropy-24-01759-f017:**
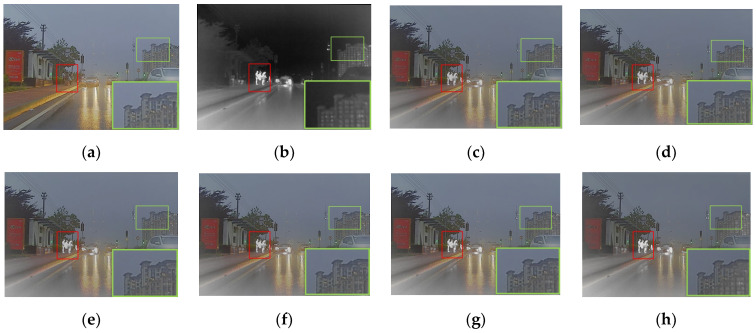
The ablation experimental results for the trade-off control parameters α and *β*. (**a**) Visible. (**b**) Infrared. (**c**) $\alpha_{1}=1$. (**d**) $\alpha_{1}=2$. (**e**) $\alpha_{1}=3$. (**f**) $\alpha_{1}=5$. (**g**) $\alpha_{1}=6$. (**h**) $\alpha_{1}=7$. (**i**) $\alpha_{1}=8$. (**j**) $\alpha_{1}=9$. (**k**) $\beta_{1}=0.1$. (**l**) $\beta_{1}=0.2$. (**m**) $\beta_{1}=0.3$. (**n**) $\beta_{1}=0.4$. (**o**) $\beta_{1}=0.5$. (**p**) $\beta_{1}=0.6$. (**q**) $\beta_{1}=0.7$. (**r**) $\beta_{1}=0.8$. (**s**) $\beta_{1}=0.9$. (**t**) Ours $\left(\alpha_{1}=4,\;\beta_{1}=0.66\right)$.

**Table 1 entropy-24-01759-t001:** Parameter settings of the feature extraction module.

Module	Layer	Input Channel	Output Channel	Activation
Shallow Feature Extraction Module	1	1	16	None
	2	16	16	Leaky ReLU
Deep Feature Extraction Module	1	16	32	Leaky ReLU
	2	32	64	Leaky ReLU

**Table 2 entropy-24-01759-t002:** Parameter settings of the image reconstruction module.

Module	Layer	Input Channel	Output Channel	Activation
Image Reconstruction Module	1	224	128	Leaky ReLU
	2	128	64	Leaky ReLU
	3	64	32	Leaky ReLU
	4	32	1	Leaky ReLU

**Table 3 entropy-24-01759-t003:** The quantitative results of the MSRS dataset. Red and blue indicate the best and the second-best results, respectively.

Metric	GTF	DenseFuse	U2Fusion	GANMcC	RFN-Nest	STDFusion	Ours
EN	5.4836	5.4013	4.2694	4.3417	5.7275	5.6072	6.0108
MI	1.4152	2.3624	1.7075	1.7122	1.8086	3.5956	4.2372
SD	6.0832	6.1874	4.7325	4.7983	6.2975	6.7386	6.9495
SF	0.0231	0.0182	0.0239	0.0248	0.0334	0.0249	0.0350
VIF	0.5170	0.6963	0.4536	0.4887	0.9274	0.6236	0.8951
Q^ab/f^	0.3361	0.3546	0.3220	0.5644	0.5874	0.4576	0.6250

**Table 4 entropy-24-01759-t004:** The quantitative results of the M^3^FD dataset. Red and blue indicate the best and the second-best results, respectively.

Metric	GTF	DenseFuse	U2Fusion	GANMcC	RFN-Nest	STDFusion	Ours
EN	6.5086	6.6316	7.0221	6.9833	6.7907	6.7776	7.1993
MI	3.0838	3.0782	3.0634	2.9526	2.6379	3.9293	4.4602
SD	9.9339	8.9730	9.3571	8.9571	8.9489	8.5261	9.0074
SF	0.0340	0.0204	0.0336	0.0362	0.0390	0.0298	0.0295
VIF	0.8978	0.8294	1.0279	1.0200	1.0047	1.0537	1.1064
Q^ab/f^	0.5001	0.3739	0.5174	0.5095	0.5048	0.5132	0.5209

**Table 5 entropy-24-01759-t005:** The quantitative results of the effectiveness of the global attention module. Red and blue indicate the best and the second-best results, respectively.

Metric	Without the 1st Attention Module	Without the 2nd Attention Module	Without Both Attention Modules	Ours
EN	6.7450	6.7584	6.6899	6.7844
MI	4.4664	4.5162	4.4048	4.7104
SD	8.9386	8.9254	8.8987	8.9747
SF	0.0320	0.0271	0.0220	0.0294
VIF	0.9936	0.9921	0.9836	1.0026
Q^ab/f^	0.5010	0.4927	0.4793	0.5199

**Table 6 entropy-24-01759-t006:** The quantitative results of the advancement of the global attention module. Red and blue indicate the best and the second-best results, respectively.

Metric	Replacing Attention Module with SENet	Replacing Attention Module with ECANet	Ours
EN	6.5809	6.6445	6.7844
MI	4.6827	4.6111	4.7104
SD	8.9298	8.9390	8.9747
SF	0.0277	0.0283	0.0294
VIF	0.9867	0.9370	1.0026
Q^ab/f^	0.5170	0.5007	0.5199

**Table 7 entropy-24-01759-t007:** The quantitative results of the effect of the number of convolution blocks. Red and blue indicate the best and the second-best results, respectively.

Metric	3-Conv-Block	4-Conv-Block	Ours (2-Conv-Block)
EN	6.7299	6.3615	6.7844
MI	4.6058	3.0175	4.7104
SD	8.9157	8.7353	8.9747
SF	0.0293	0.0296	0.0294
VIF	0.9843	0.8739	1.0026
Q^ab/f^	0.5310	0.4540	0.5199

**Table 8 entropy-24-01759-t008:** The ablation experimental results for the trade-off control parameter α. Red and blue indicate the best and the second-best results, respectively.

Metric	$$\alpha_{1}=1$$	$$\alpha_{1}=2$$	$$\alpha_{1}=3$$	$$\alpha_{1}=5$$	$$\alpha_{1}=6$$	$$\alpha_{1}=7$$	$$\alpha_{1}=8$$	$$\alpha_{1}=9$$	Ours $$\left(\alpha_{1}=4\right)$$
EN	6.5890	6.6176	6.6162	6.6245	6.6959	6.6796	6.7081	6.6526	6.7844
MI	4.2757	4.4161	4.4141	4.4687	4.5292	4.5340	3.9771	3.6744	4.7104
SD	8.6407	8.6807	8.6769	8.6910	8.7493	8.5791	8.7539	8.7759	8.9747
SF	0.0289	0.0289	0.0293	0.0269	0.0271	0.0257	0.0218	0.0179	0.0294
VIF	0.9412	0.9467	0.9455	0.9462	0.9625	0.9423	0.9358	0.9341	1.0026
Q^ab/f^	0.5290	0.5287	0.5240	0.5082	0.4921	0.4509	0.4201	0.3629	0.5199

**Table 9 entropy-24-01759-t009:** The ablation experimental results for the trade-off control parameter *β*. Red and blue indicate the best and the second-best results, respectively.

Metric	$\beta_{1}=0.1$	$\beta_{1}=0.2$	$\beta_{1}=0.3$	$\beta_{1}=0.4$	$\beta_{1}=0.5$	$\beta_{1}=0.6$	$\beta_{1}=0.7$	$\beta_{1}=0.8$	$\beta_{1}=0.9$	Ours $\left(\beta_{1}=0.66\right)$
EN	6.2031	6.3875	6.5618	6.6762	6.7122	6.7530	6.7006	6.7238	6.7444	6.7844
MI	3.1523	3.5214	3.8086	4.7037	4.5235	4.6019	4.6483	4.4814	4.3923	4.7104
SD	8.3903	8.8980	8.8988	8.9110	8.9037	8.9236	8.9331	8.9339	8.9419	8.9747
SF	0.0296	0.0283	0.0292	0.0298	0.0286	0.0288	0.0284	0.0285	0.0263	0.0294
VIF	0.8754	0.8825	0.9089	0.9730	0.9754	0.9915	0.9735	0.9819	0.9807	1.0026
Q^ab/f^	0.5133	0.5124	0.5275	0.5350	0.5308	0.5261	0.5187	0.5154	0.4890	0.5199

**Table 10 entropy-24-01759-t010:** The ablation experimental results for the trade-off control parameter *β*. Red and blue indicate the best and the second-best results, respectively.

Metric	$\beta_{1}=0.61$	$\beta_{1}=0.62$	$\beta_{1}=0.63$	$\beta_{1}=0.64$	$\beta_{1}=0.65$	$\beta_{1}=0.67$	$\beta_{1}=0.68$	$\beta_{1}=0.69$	Ours $\left(\beta_{1}=0.66\right)$
EN	6.7816	6.7521	6.7838	6.7537	6.7061	6.7635	6.7244	6.7521	6.7844
MI	4.8015	4.5154	4.8101	4.6564	4.5619	4.5265	4.5265	4.7309	4.7104
SD	8.9698	8.9504	8.9545	8.9312	8.9061	8.9546	8.9399	8.9504	8.9747
SF	0.0282	0.0290	0.0283	0.0288	0.0291	0.0280	0.0288	0.0290	0.0294
VIF	1.0003	0.9940	0.9946	1.0012	0.9788	0.9944	0.9861	0.9940	1.0026
Q^ab/f^	0.5170	0.5248	0.5235	0.5138	0.5264	0.5111	0.5250	0.5248	0.5199

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
