# Peer review of "Infrared and Visible Image Fusion for Highlighting Salient Targets in the Night Scene"

_entropy, 2022, doi:10.3390/e24121759_

Round 1

Reviewer 1 Report

Comments

1.    The comparison of ablation in Fig.13 is not clear to see the differences

2.     The Introduction has a repetitive description of the implementation of the paper and lacks details on the application of fusion images.

3.     The layout of figures and formulas needs to be adjusted and polished.

4. In terms of the model, the present network model does not have a bunch of convolutional layers compared to other networks, which can avoid the high computational overhead.

Questions: 

1. Can we continue to add a deeper feature extraction and an attention module to improve performance?

2. Can trade-off control parameters continue to be tuned to improve performance?

 3. Is it possible to connect the Global Attention Module of the visible and infrared ima processing to better assign weights to the channels?

Author Response

Thank you very much for the comments and suggestions provided by the three reviewers. Based on all the comments and suggestions, I have revised this paper. The details are as follows.

Apply to the Reviewer 1

  1. The comparison of ablation in Fig.13 is not clear to see the differences.

Response: Thanks for pointing out this problem. In Figure 13, there is no denying that all global attention modules can effectively highlight the salient targets. So, it is difficult to judge the superiority of these modules by human visual. Therefore, we selected 100 pairs of images for further quantitative analysis. As shown in Table 6, all evaluation metrics of the proposed module achieve maximum values. Therefore, the above results can confirm the advancement of the proposed attention module.

  1. The Introduction has a repetitive description of the implementation of the paper and lacks details on the application of fusion images.

Response: Thanks for pointing out this problem. We deleted the repetitive description of the implementation of the paper, and then added the application of fusion images in the first paragraph of the introduction (in lines 24-27).

  1. The layout of figures and formulas needs to be adjusted and polished.

Response: Thanks for pointing out this problem. We optimized the English expression and adjusted the layout of figures and formulas in the paper.

  1. In terms of the model, the present network model does not have a bunch of convolutional layers compared to other networks, which can avoid the high computational overhead.

Response: Thanks for your comment.

  1. Can we continue to add a deeper feature extraction and an attention module to improve performance?

Response: Thanks for pointing out this problem. In Chapter 4, we added a set of ablation experiments concerning the feature extraction module (in lines 403-417). Three or four convolution blocks are adopted in the deep feature extraction module. One hundred pairs of images on the M3FD dataset are chosen for quantitative analysis. As shown in Table 7, the proposed method achieves the maximum value on 4 evaluation metrics, including EN, MI, SD and VIF. Noting that the evaluation metrics do not rise with the increase of convolution blocks, which means that the network may be overfitted. Therefore, only two convolution blocks are adopted in deep feature extraction module.

  1. Can trade-off control parameters continue to be tuned to improve performance?

Response: Thanks for pointing out this problem. In Chapter 4, we added a set of ablation experiments to further verify the validity of the luminance estimation function (in lines 447-467). We set and keep other parameters in the network unchanged.

  1. Is it possible to connect the Global Attention Module of the visible and infrared ima processing to better assign weights to the channels?

Response: Thanks for pointing out this problem. After experimental analysis, we modified the network structure. The new network structure of the proposed method is shown in Figure 4.

Apply to the Reviewer 2

  1. Please add citations at locations where the state-of-the-art method is referenced in the paper.

Response: Thanks for pointing out this problem. We added citations at the corresponding position (in lines 262-264,274-277).

  1. The authors are also suggested to provide some statistical results to demonstrate if the improvements are statistically significant. 

Response: Thanks for pointing out this problem. First, we added two evaluation metrics (in lines 262-264). In addition, we increased the number of the images for quantitative comparison in the ablation experiment to improve the statistical significance of the experiment.

  1. Some related references about feature fusion should be added in the Introduction Section, such as A. Zhang, Z. Min, Z. Zhang and M. Q. . -H. Meng, "Generalized Point Set Registration With Fuzzy Correspondences Based on Variational Bayesian Inference," in IEEE Transactions on Fuzzy Systems, vol. 30, no. 6, pp. 1529-1540, June 2022, doi: 10.1109/TFUZZ.2022.3159099.

Response: Thanks for offering this advice. We added this paper as the fourth reference in the Introduction Section.

Apply to the Reviewer 3

The paper presents a method for the fusion of nighttime images acquired in the visible and thermal infrared bands. The method is first introduced against the state of the art, then detailed and finally validated (qualitatively and quantitatively) with existing datasets in comparison with other methods. Within the validation phase, also an ablation experiment is proposed, in the parameters of the algorithm are varied to show the quality of the results. I think the paper could be of interest for the readers of Entropy. The manuscript is well-written and the experimental passages are clear.

Response: Thanks for your comments. We optimized the English expression and adjusted the layout of figures and formulas in the paper.

Reviewer 2 Report

1. Please add citations at locations where the state-of-the-art method is referenced in the paper.

2. The authors are also suggested to provide some statistical results to demonstrate if the improvements are statistically significant. 

3. Some related references about feature fusion should be added in the Introduction Section, such as 

A. Zhang, Z. Min, Z. Zhang and M. Q. . -H. Meng, "Generalized Point Set Registration With Fuzzy Correspondences Based on Variational Bayesian Inference," in IEEE Transactions on Fuzzy Systems, vol. 30, no. 6, pp. 1529-1540, June 2022, doi: 10.1109/TFUZZ.2022.3159099.

Author Response

(The authors gave the same response as above.)

Reviewer 3 Report

The paper presents a method for the fusion of nighttime images acquired in the visible and themal infrared bands. The method is first introduced against the state of the art, then detailed and finally validated (qualitatively and quantitatively) with existing datasets in comparison with other methods. Within the validation phase, also an ablation experiment is proposed, in the parameters of the algorithm are varied to show the quality of the results.
I think the paper could be of interest for the readers of Entropy. The manuscript is well-written and the experimental passages are clear.

Author Response

(The authors gave the same response as above.)
